# The Prevalence of Onchocerciasis-Associated Epilepsy in Mundri West and East Counties, South Sudan: A Door-to-Door Survey

**DOI:** 10.3390/pathogens11040396

**Published:** 2022-03-24

**Authors:** Stephen Raimon Jada, Alfred Dusabimana, Gasim Abd-Elfarag, Samuel Okaro, Nele Brusselaers, Jane Y. Carter, Makoy Yibi Logora, Jacopo Mattia Rovarini, Charles R. Newton, Robert Colebunders

**Affiliations:** 1Amref Health Africa, Juba P.O. Box 410, South Sudan; stephenraimon@gmail.com (S.R.J.); samuel.okaro@amref.org (S.O.); jacopo.rovarini@amref.it (J.M.R.); 2Global Health Institute, University of Antwerp, 2016 Antwerp, Belgium; alfred.dusabimana@gmail.com (A.D.); nele.brusselaers@uantwerpen.be (N.B.); 3Amsterdam Center for Global Health, Department of Paediatrics and Department of Global Health, Amsterdam UMC, 1105 AZ Amsterdam, The Netherlands; gasim4u83@gmail.com; 4Amsterdam Institute for Global Health and Development, 1105 AZ Amsterdam, The Netherlands; 5Centre for Translational Microbiome Research, Karolinska Institutet, 17177 Stockholm, Sweden; 6Amref Health Africa Headquarters, Nairobi P.O. Box 30125, Kenya; jane.carter@amref.org; 7Neglected Tropical Diseases Unit, Ministry of Health, Juba P.O. Box 410, South Sudan; morrelogora@yahoo.com; 8Department of Psychiatry, University of Oxford, Oxford OX1 2JD, UK; charles.newton@psych.ox.ac.uk

**Keywords:** onchocerciasis, *Onchocerca volvulus*, epilepsy, nodding syndrome, ivermectin

## Abstract

A two-phase survey of epilepsy was conducted in selected villages in Mundri West and East Counties (26 June–8 July, 2021), an onchocerciasis-endemic area in Western Equatoria State in South Sudan. In the first phase, households were visited by a trained research team to identify persons suspected to have epilepsy. In the second phase, persons suspected to have epilepsy were interviewed and examined by a clinician to confirm the diagnosis. A total of 364 households agreed to participate in the survey, amounting to 2588 individuals. The epilepsy screening questionnaire identified 91 (3.5%) persons with suspected epilepsy, of whom the diagnosis of epilepsy was confirmed by a clinician in 86 (94.5%). The overall prevalence of confirmed epilepsy was 3.3% (95% CI: 2.7–4.1%), and of nodding syndrome was 0.9% (95% CI: 0.6–1.4%). In 61 (16.8%) households there was at least one person with epilepsy. Only 1212 (46.9%) of 2583 people took ivermectin during the last distribution round in 2021. The annual epilepsy incidence was 77.3/100,000 (95% CI: 9.4–278.9/100,000) and the annual epilepsy mortality was 251.2/100,000 (95% CI: 133.8–428.7/100,000). In conclusion, a high prevalence and incidence of epilepsy was observed in villages in Mundri. Urgent action is needed to prevent children from developing onchocerciasis-associated epilepsy by strengthening the local onchocerciasis-elimination programme.

## 1. Introduction

A high prevalence of epilepsy is observed in many onchocerciasis-endemic areas, where persons with nodding syndrome and Nakalanga syndrome are reported [1]. Recent studies have shown that nodding syndrome and Nakalanga features are phenotypic presentations of onchocerciasis-associated epilepsy (OAE) [1]. Indeed, children with nodding syndrome, and others meeting OAE criteria without nodding syndrome, live in the same families and in villages located close to rapidly flowing rivers with blackfly breeding sites [1]. A recent post-mortem study showed that the brains of persons who had died with nodding syndrome or another form of OAE had similar pathological findings [2]. Both nodding syndrome and other forms of OAE appear in areas with high *Onchocerca volvulus* transmission, and stop appearing when onchocerciasis is eliminated [3,4]. 

Between 2001 and 2002, three small case–control studies were performed in Greater Mundri in Western Equatoria, South Sudan, to identify the potential risk factors for developing nodding syndrome [5]. These studies, performed in Mundri Centre, Lui, and Amadi Payams—three administrative divisions close to the Yei River—showed a strong positive association between nodding syndrome and onchocerciasis. Indeed, *O. volvulus* microfilariae were observed in 92.3% of skin snips from epilepsy cases compared with 43.7% of controls (Odds Ratio (OR) 15.4; 95% Confidence Interval (CI): 1.60–148.82 (*p* = 0.008)) [5]. At that time, the prevalence of nodding syndrome was estimated at 2.3% (41/1783) in Lui and 6.7% (57/854) in Amadi, with the first cases of nodding syndrome estimated to have appeared in Mundri in the 1990s [5]. 

In 2018, in a door-to-door survey performed in onchocerciasis-endemic villages in Maridi, Western Equatoria state, South Sudan, an epilepsy prevalence of 4.4% was documented with a prevalence of 11.9% in a village close to the Maridi dam, the only known blackfly breeding site in the area [6]. In this survey, 85.2% of persons with epilepsy (PWE) met the criteria of OAE, and 45.5% met the criteria of nodding syndrome [7]. In June 2020 a similar survey in Mvolo County documented an epilepsy prevalence of 5.1%, whereby 78.4% of PWE met the criteria of OAE and 44.1% of PWE had a history of nodding seizures [8]. 

Recent cross-sectional studies suggest that strengthening an onchocerciasis-elimination programme leads to a reduction in the incidence of nodding syndrome and OAE [3,4]. However, this has not yet been demonstrated in a prospective study, and it is also not known which onchocerciasis-elimination measures are most effective. To investigate these questions, we initiated a prospective study; it compared the effect of bi-annual community-directed treatment with ivermectin (CDTI) together with community-based vector control using a “slash and clear” method in Maridi [9], with annual CDTI in Mvolo and Mundri, on the incidence of nodding syndrome and other forms of epilepsy [10]. As part of this initiative, a baseline epilepsy survey was completed in Maridi [6] and Mvolo [8]. In this paper we present the results of a baseline survey carried out in the Greater Mundri area. 

## 2. Methodology

### 2.1. Study Setting

The study was conducted in Mundri West and East Counties in Western Equatoria State of South Sudan (Figure 1).

An epilepsy survey was conducted in selected villages in Amadi and Mundri Payams (Mundri West County) and Lui Payam (Mundri East County) between 26 June−8 July, 2021, in Western Equatoria State, South Sudan. Villages, with the exception of Lui town, were chosen because of their proximity to the Yei River (Figure 2).

The projected population of the Greater Mundri area is estimated at 115,717 (47,853 in Mundri West County; 67,864 in Mundri East County). Moru is the dominant ethnic group in the Greater Mundri area; other ethnic groups include Moru-Kodo and Avukaya. Moru is the dominant language in both Mundri East and Mundri West, but the common language of communication is local Arabic.

Farming is the main economic and livelihood activity; the main crops grown in Mundri are sorghum, *simsim* (sesame), cassava, sweet potatoes, groundnuts, vegetables, maize and millet. Fishing is also a viable supplementary livelihood activity in Mundri West. Water is mainly sourced from the Yei and Mori Rivers, which are fast-flowing with several shallow rapids. Goats and chickens are the main livestock in Mundri West and Mundri East. No pigs are kept in the area.

### 2.2. Study Design

A door-to-door survey was conducted in seven selected villages in Mundri West and East Counties. A two-step survey was used to identify PWE. In the first phase, households were visited by a trained research team of ten locally recruited research assistants (each household was visited by a single research assistant). After obtaining informed consent, family members were interviewed using a validated questionnaire translated into the local language [11]. In the second phase, persons with suspected epilepsy identified by the research assistants were referred for interview and examination by five clinical officers trained to diagnose epilepsy, according to International League Against Epilepsy (ILAE) definitions [12]. These clinical officers confirmed the diagnosis of epilepsy or suggested an alternative diagnosis. The ten research assistants and five clinical officers were selected by the local health authorities from the Mundri West and Mundri East County Health Departments. Research assistants were selected from among county community drug distributors who had at least a secondary school education level. The research assistants were trained for one full day on the use of the screening questionnaire for suspected epilepsy. In addition, five clinical officers from primary healthcare centres in the counties were trained on how to confirm the diagnosis of epilepsy. Both the research assistants and clinical officers pilot tested the data collection tools prior to data collection. The training was organised by two medical doctors (SR and GA). During the home visits, SR supervised the research assistants and clinical officers, and also interviewed and examined selected individuals with suspected epilepsy.

### 2.3. Definitions

In accordance with the ILAE, a case of epilepsy was defined as an individual with at least two unprovoked seizures with at least 24 h separating the two episodes [13].

Nodding seizures were defined as the head dropping forward repeatedly in a person during a brief period of reduced consciousness. Nodding syndrome was defined according to the World Health Organisation consensus definition [14]. OAE was defined as a person meeting all the following six criteria: (i) a history of at least two unprovoked epileptic seizures at least 24 h apart; (ii) living for at least 3 years in an onchocerciasis-endemic region; (iii) living in a village with a high number of persons with epilepsy (epilepsy well known by the community) and with families with more than one child with epilepsy; (iv) no other obvious cause of epilepsy as determined by the history; (v) onset of epilepsy between the ages of 3 and 18 years; and (vi) normal psychomotor development before the onset of epilepsy as reported by the parents [1]. 

As potential “obvious causes of epilepsy” we considered a history of a perinatal event, severe malaria, encephalitis or meningitis, and head injury with loss of consciousness in the 5 years preceding the onset of epileptic seizures. Nakalanga features were defined as an association with growth retardation without obvious cause, delay or absence of external signs of secondary sexual development, severe learning difficulties, epilepsy, and often facial, thoracic, and spinal abnormalities [15]. A “permanent household” was defined as a family who had lived in the village for at least 20 years; an “immigrant household” referred to a family who had lived in the village for less than 20 years.

### 2.4. Data Collection, Management 

Screening of households by research assistants to identify persons with suspected epilepsy was performed using a paper-based questionnaire (Appendix A) addressed to each family member. If nobody or only children were present in a house, this house was revisited at least one further time. For every household member, whether a person was present or absent was noted. If a person was absent, information about him/her was obtained from the other household members. This questionnaire included five epilepsy screening questions adapted from a questionnaire previously validated in Mauritania [11]. After these five questions had been used during an epilepsy survey in Maridi, South Sudan [6], we decided to remove the question “sudden onset of brief body sensations, hallucinations or illusions be they visual, auditory or olfactory” from the previously validated questionnaire; this was because we experienced that this question was difficult to translate into the local language and was poorly understood, and it was exceptional that a person with confirmed epilepsy would be identified only by a positive answer to this question. We replaced this question with a question about head nodding episodes. If the answer to one of the five epilepsy questions was positive, this person was considered to have suspected epilepsy and was referred to one of the clinicians. Each household member was also asked whether he/she experienced skin itching and/or problems with vision (blindness or blurred vision), and whether they had taken the drug ivermectin during the previous distribution in 2021. The questionnaire also contained questions about duration of residence, ethnicity, main income-generating activity of the family, exposure to cattle or pigs, whether family members had recently developed epilepsy, and whether family members with epilepsy had recently died (when and at what age). For confirmation of epilepsy, clinical officers and medical doctors used a questionnaire deployed on the KoBo toolbox, a free open-source tool for mobile data collection (https://kobo.humanitarianresponse.info/accounts/login/?next=%2F%23%2F#/; accessed on 21 March 2022), with unique codes assigned to each suspected case. Clinicians assessed the type of epilepsy, triggers of seizures, causes of epilepsy, epilepsy-related co-morbidities (cognitive impairment, behavioural problems, burn scars). Cognitive function was evaluated by determining whether the participant was well oriented in time and space, and whether he/she could remember his/her name, was coherent in speech, and was obedient to orders. PWE were also examined for onchocerciasis-related clinical signs (skin lesions, vision problems). Blurred vision or blindness was considered if a person was unable to see the five fingers of his hand. Level of autonomy was assessed using a modified Rankin scale of 1−5 [16]. Questions were asked about current and past anti-seizure medication and ivermectin intake in 2021. 

### 2.5. Onchocerciasis Antibody Testing of Children 

A total of 224 children aged between 3–9 years, living in Amadi and Mundri Centre, were tested for IgG antibodies to *O. volvulus* using the Ov16 rapid diagnostic test (Standard Diagnostics, Inc., Yongin-si, South Korea) as an indicator of the degree of recent *O. volvulus* transmission. Blood was obtained via finger prick and the test was performed according to the manufacturer’s instructions. Among these Ov16-tested children, we determined ivermectin use during the last CDTI round in 2021.

### 2.6. Data Analysis

Continuous variables were summarised with median and interquartile range (IQR), while the absolute and relative frequencies were used to summarise categorical variables. A Chi-squared Exact test was used to compare categorical variables, and Wilcoxon signed-rank tests were used for continuous variables. The epilepsy prevalence in a village was calculated by summing up all the epilepsy confirmed cases in the village and dividing them by the total number of persons residing in that village who had participated in the survey. The corresponding 95% Clopper–Pearson confidence interval (exact interval) was used. This interval was preferred to the Wald confidence interval because it is based on the exact binomial distribution rather than the normal approximation of the Wald confidence interval. The overall epilepsy incidence in the total population was calculated by summing up all cases of epilepsy in which seizures started within the previous 5 years (epilepsy duration between 0 and 5 years), dividing by the total population at risk during this period, and dividing by five to obtain the annual incidence. Epilepsy-related mortality was calculated by dividing the number of persons known to have epilepsy who had died in the previous 12 months by the summed person-years of the population at risk over this period. Ivermectin coverage was defined as the percentage of the population that reported taking ivermectin in 2021. A two-level hierarchical generalised linear mixed model (GLMM) with logit link was considered to assess variables associated with epilepsy. A household was considered as the first hierarchical level to account for the similarity among individuals living in the same house, and a village was considered as the second hierarchical level to account for the similarity among individuals residing in the same village. Adjusted odds ratios with 95% confidence intervals (CI) in multivariable analysis were determined. All two-way interactions between variables were considered in the model, and a likelihood ratio test was used to identify potential interactions. 

## 3. Results

### 3.1. Households Participating in the Survey

In total, 364 households containing 2,588 individuals were visited, of whom 526 (20.3%) were not present at the time of the interview; however, the interviewer obtained information about them from their family members. The median number of individuals living in the same house was four (IQR: 2−6), with a maximum of 20 individuals living in the same house. Three hundred and forty-six (95.1%) families belonged to the Moro ethnic group, and 18 (4.9%) families belonged to other ethnic groups including Nyamosa and Pojulu. Reported income-generating activities included farming in 305 (83.8%) families, fishing in 50 (13.7%), cattle keeping in 19 (5.5%), trading in 18 (4.9%), and working for the local government in 102 (28.0%). Two hundred and ninety-four (80.8%) families originated from the selected villages, while 70 (19.2%) were immigrant families with a median (IQR) period of residence in the village of seven (3.0−10.0) years. 

### 3.2. Prevalence of Epilepsy and Potential Onchocerciasis-Associated Co-Morbidities

Of the 2588 individuals included in the survey, 91 (3.5%) were identified in the screening questionnaire as persons with suspected epilepsy and were referred to the clinician. In 86 individuals (95.5%) the diagnosis of epilepsy was confirmed by the clinical officer and/or a medical doctor (SR). Five persons with suspected epilepsy were diagnosed with another condition: severe malaria (n = 1), temporary amnesia (n = 1), paroxysmal vertigo (n = 1), recurrent febrile convulsions (n = 1), and psychogenic seizures (n = 1) (Figure 3). The overall prevalence of confirmed epilepsy was 3.3%, (95% CI: 2.7–4.1%). Itching was reported in 1041 (40.2%) individuals, and blindness or blurred vision in at least one eye in 70 (2.7%) (Table 1). The highest number of PWE was observed in Hai Lenderwa, at 10 (6.6%), and Hai Gabat village, at 14 (4.3%).

In 61 (16.8%) of 364 households who participated in the survey, there was at least one household member with epilepsy; in 19 (5.2%) of 364 households, there were two PWE; and in one household, there were three PWE.

### 3.3. Characteristics of Persons with Epilepsy

Of the 86 PWE, 43 (50.0%) were female (Table 2). The median age of PWE was 25.0 (20.0–29.0) years. Of the 81 PWE for which the age of onset of seizures was known, the first seizures most frequently appeared between the ages of 5–15 years (Figure 4).

The median age of onset of seizures was 10.0 (IQR 8.0–15.0) years. The most frequent seizure type was generalised convulsive seizures in 61 (72.6%). Twenty-three (26.7%) of the 86 PWE met the criteria of probable nodding syndrome, including seven (8.1%) who only experienced nodding seizures and 16 (18.6%) who experienced both nodding and convulsive seizures. In three individuals (3.5%) there was a history of blurred vision or blindness in at least one eye. In 67 (79.8%) PWE, there was no specific trigger that provoked the seizures; however, in eight (9.5%), seizures were precipitated by the sight of food, and in nine (10.7%), by cold weather. According to family members, six (7.1%) PWE were severely cognitively impaired. Papular/nodular pruritic lesions were present in 12 (14.3%) PWE; burn scars in 14 (16.7); and itching was reported in 20 (23.8%). Overall, 65 (75.6%) of 86 PWE met the OAE criteria. Two (2.5%) PWE presented with Nakalanga features. Eighty-two (95.4%) PWE had been treated with anti-seizure medication and were currently taking the medication, mainly carbamazepine, which was taken by 55 (67.1%). Fifty-eight (67.4%) PWE reported having ever taken ivermectin, but only 39.3% had taken ivermectin in 2021. In 45 PWE, there was a family history of epilepsy, and in 37 PWE (97.5%), this was a sibling (Table 2).

Epilepsy was most frequent in the age group 21−25 years at 25/247 (10.1%) (Figure 5).

Twenty-three PWE were diagnosed with probable nodding syndrome. The overall prevalence of probable nodding syndrome was therefore 0.9%. Persons with nodding syndrome were younger, presented their first seizures at a younger age, and their seizures were more often triggered by the sight of food or cold compared to persons with other forms of epilepsy (Table 3).

### 3.4. Incidence of Epilepsy

Two PWE developed their first seizures in the 12 months preceding the survey (annual incidence of 77.3/100,000, 95% CI: 9.4–278.9/100,000) and four PWE developed their first seizures in the previous 5 years, corresponding to an incidence of 154.6/100,000, 95% CI: 42.1–395.3/100,000 per 5 years.

### 3.5. Onchocerciasis-Related Morbidity in the Family

In 21 (5.8%) households, there was at least one person with a history of nodding syndrome or nodding syndrome plus other seizure types. In 244 (67.0%) households, there was at least one person with skin itching, and in 61 (17.0%) households there was at least one person with blindness or blurred vision, with eight (2.2%) families with more than one person with blindness or blurred vision. The median age of persons with skin itching was 18.0, (IQR: 10.0–30.0) and with blindness or blurred vision was 54.5, (IQR:40.0–66.0) years.

### 3.6. Mortality of Persons with Suspected Epilepsy

Overall, 41 persons died during the previous 24 months; of these, 13 (36.6%) were persons with epilepsy according to the family. This corresponds to 251.2/100,000 (95% CI: 133.9–428.7/100,000) deaths of persons suspected to have epilepsy per year. The median age of death of people with suspected epilepsy was 22.0 (IQR 15.0−35.0) years, in the range of 20−35 years.

### 3.7. Risk Factors for Epilepsy

The variance of the random effect attributed to the village was zero. Therefore, we only considered family as the random effect in the GLMM model. This model revealed that the probability of living with epilepsy increased with increasing age up to 29.9 years, but decreased above this age (Table 4 and Figure 6). Moreover, the probability of living with epilepsy was high among people living in a family with a household member with blindness or blurred vision in at least one eye, but the association was insignificant (*p*-value = 0.067).

Including all the villages in the analysis, distance to the Yei River was not found to be a risk factor for epilepsy. However, excluding Lui town—the site where the epilepsy treatment centre is located—from the analysis, a higher prevalence of epilepsy was observed in villages very close to the Yei River (Hai Gabat, Hai Ngulawa, Hai Lenderwa) compared to those located further away from the Yei River (Hai Malakia, Hai Facki, Hai Mirikalanga) (*p*-value = 0.041) (Table 5).

A proportion of 41.8% of the PWE had taken ivermectin in 2021, and this proportion was comparable to the overall study population (46.9%) (Table 6).

In total, 224 children (74 from Hai Gabat village in Amadi Payam and 150 from Hai Milikanga, Hai Ngolawa and Hai Matara village in Mundri Centre Payam) aged between 3−9 years were tested for IgG antibodies to *O. volvulus* using the Ov16 rapid diagnostic test, to evaluate the degree of onchocerciasis transmission. Of those children eligible for ivermectin (5 years and above), only 22/62 (35.5%) in Amadi and 21/109 (19.3%) in Mundri Centre had taken ivermectin during the last CDTI round (Table 7).

## 4. Discussion

Our study confirms a high prevalence of epilepsy (3.3%) in villages in Mundri East and West Counties in Western Equatoria State, South Sudan. This prevalence is slightly lower than the 4.4% epilepsy prevalence in Maridi [6] and the 5.1% prevalence in Mvolo. A very high percentage (97.8%) of persons with suspected epilepsy, based on answers to the five epilepsy screening questions [11], were confirmed to have epilepsy. This has also been observed in other areas with high epilepsy prevalence, where the affected population is generally well informed about epilepsy [6,17], in contrast to areas of low epilepsy prevalence [18]. In Maridi, the diagnosis of epilepsy was confirmed in 92.1% of persons with suspected epilepsy. The higher percentage of confirmed cases in Mundri is explained by the fact that in Mundri, we did not include the screening question “sudden onset of brief body sensations, hallucinations or illusions be they visual, auditory or olfactory”. This question was previously noted as having the lowest positive predictive value for the diagnosis of epilepsy [6]; in only one of the suspected epilepsy cases was a diagnosis of febrile seizures made. The reason for more children with febrile seizures not being considered as suspected persons with epilepsy was that the local population of Mundri generally considers febrile seizures as malaria, and not as potential epilepsy. This, however, may have led to an underestimation of the number of epilepsy cases in the <5 years age group.

Of the PWE, 75.6% met the OAE criteria, and in 26.7% of PWE, there was a history of nodding seizures. These percentages were lower than in Maridi, where 91.8% of PWE met the OAE criteria and 45.5% were classified as having probable nodding syndrome [7]. The overall prevalence of probable nodding syndrome (0.9%) is lower than the previously reported 2.3% prevalence in Lui town and 6.7% in Amadi [5]. It may be that there has been a decrease in the incidence of nodding syndrome because of the re-introduction of CDTI since 2017. However, given the low therapeutic coverage of ivermectin, it is likely that certain persons with other forms of epilepsy have previously been considered as having nodding syndrome.

The highest epilepsy prevalence was observed in the 20–25 year age group. This is in contrast with the findings from similar surveys in Maridi and Mvolo, where the highest epilepsy prevalence was observed in the 10–20 year age group. In all of these three sites, the ivermectin coverage was low; however, the survey in Mundri was conducted in 2021 after four years of annual CDTI, while in Maridi, the survey was conducted in 2018 after only one CDTI in 2017 [6]. Moreover, in Mundri, the lower OV16 seroprevalence among children below the age of 10 years suggests a lower level of onchocerciasis transmission. This also explains the lower annual incidence of epilepsy in Mundri (77.3/100,000, 95% CI: 21.2–281.3/100,000) compared with the annual incidence in Maridi (373.9/100,000) [6].

The probability of epilepsy increased with increasing age up to 29.9 years and decreased thereafter. This is explained by the high mortality of persons with OAE below the age of 30 years. Indeed, the median age of death of people with suspected epilepsy was 22 years. A very high proportion of recent deaths (36.6%) reported by families occurred in persons considered to have epilepsy. The 231.8/100,000 annual mortality of persons with suspected epilepsy in Mundri was similar to the 294.6/100,000 reported mortality in Maridi [6]. A large proportion of our study population reported itching (40.2%), and in 2.7%, blindness or blurred vision of at least one eye was reported. These findings suggest a very high level of past and ongoing *O. volvulus* transmission, explained by the low coverage of CDTI (46.9%). This percentage is much lower than the coverage percentages reported by the community-directed distributors (84.7% in Mundri West and 80.2% Mundri East), and the coverage percentages calculated using the National Bureau of Statistics projected population as denominators (83.4% in Mundri West and 70.2% in Mundri East) [19]. However, we were unable to compare our ivermectin coverage data with the coverage of data reported by community-directed distributors per study village.

More PWE were observed in families with a member who was blind or had blurred vision. This suggests that both epilepsy and blindness may have a common aetiology. In contrast to the findings from Maridi, belonging to a farming family was not a risk factor for epilepsy [6]. Including all the study villages in the analysis, in contrast to Maridi [6] and Mvolo [8], living in a village close to a river was not found to be a risk factor for developing epilepsy. However, excluding Lui town from the analysis, living close to the river was also found to be a risk for developing epilepsy. The high prevalence of epilepsy in Lui town, despite its location more than 7km from the Yei River, is most likely explained by the fact that the only epilepsy treatment centre in the region is located in Lui hospital. This may have attracted families with members affected by epilepsy to live in Lui town. Indeed, during the survey in Lui town, several PWE reported that they were not originally from Lui town and that they presented their first seizures in another village [20].

The OV16 seroprevalence in children below the age of 10 years was higher in Amadi than in Mundri Centre, suggesting higher ongoing *O. volvulus* transmission in areas closer to the river. The OV16 seropositivity of the 3-year-old children may be attributed to the presence of *O. volvulus* antibodies acquired from the mother. However, the presence of such antibodies in 5–9-year-old children is indicative of ongoing *O. volvulus* transmission in the area.

The strength of our study is that this was a population-based epilepsy survey in which all households were visited by the research team, and information was obtained on each household member concerning potential onchocerciasis-related symptoms and signs, and ivermectin use. However, there were also several limitations of the study. First, our study villages are not representative of all villages in the Greater Mundri area because we selected those located close to the Yei River. Similar to several other studies in onchocerciasis endemic areas [1,7,8,21], the distance of a village to the river was associated with a higher prevalence of epilepsy. However, because of a lack of tablets, we were unable to collect GPS information on all the households included in the study. We also lacked information about the localisation of blackfly breeding sites and blackfly biting rates in the study villages. Moreover, the diagnoses of epilepsy were not confirmed by a neurologist and no EEGs were performed. No laboratory studies or imaging investigations were performed to further investigate the causes of epilepsy. However, as there are no pigs in the Mundri area, neurocysticercosis cannot explain the high epilepsy prevalence. Clinicians only examined persons with suspected epilepsy referred by the research assistants and did not visit households where no PWE were reported. However, given the very high epilepsy prevalence, we do not expect that additional PWE were missed.

A large percentage (20.3%) of the study population was not present during the home visits, and information about them was only obtained from family members. However, as most persons away from their homes were working in the fields, they were likely to be healthier than those who were present at home. The diagnosis of cognitive impairment, behavioural problems, and blindness were only assessed by interview. It is important to note is that none of the children with epilepsy were below the age of five years, and no deaths of children with epilepsy were reported in this age group. This is in contrast with population-based epilepsy surveys in non-onchocerciasis endemic regions in Africa, where perinatal anoxia and cerebral palsy are important causes of epilepsy in this age group [22,23]. It may be that in some children below the age of five years, epileptic seizures were considered as febrile seizures. Given the cross-sectional study design, epilepsy-related mortality could only be estimated retrospectively by interview and the information obtained could have been influenced by recall bias. We only obtained information from families on whether a family member with epilepsy had died in the previous year, but we do not know whether the diagnosis of epilepsy was confirmed by a medical doctor, and no verbal autopsies were performed.

## 5. Conclusions

This study confirms the high prevalence of epilepsy, including nodding syndrome, in onchocerciasis-endemic areas where the onchocerciasis-elimination programme is functioning sub-optimally. The ivermectin coverage in villages in Mundri in 2021 was very low (47.4%). Ideally, bi-annual distribution of ivermectin—as was successfully implemented in northern Uganda [3]—is needed, to reduce the incidence of epilepsy and interrupt the development of OAE. We also propose an entomological study in the Greater Mundri area to investigate whether, as in Maridi County, a community based “slash and clear” vector control method could be implemented. Urgent international solidarity is needed to protect children in Mundri from developing epilepsy and other irreversible neurological abnormalities, and to provide uninterrupted anti-seizure medication to all those affected by epilepsy.

## Figures and Tables

**Figure 1 pathogens-11-00396-f001:**
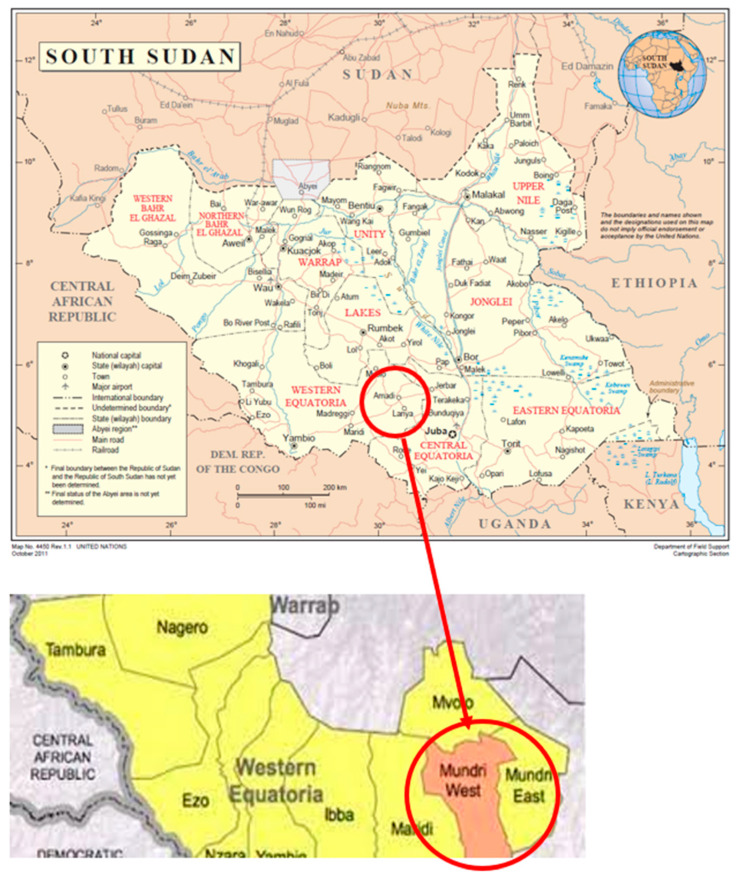
Map of South Sudan and map of Western Equatoria State with Mundri West and Mundri East Counties.

**Figure 2 pathogens-11-00396-f002:**
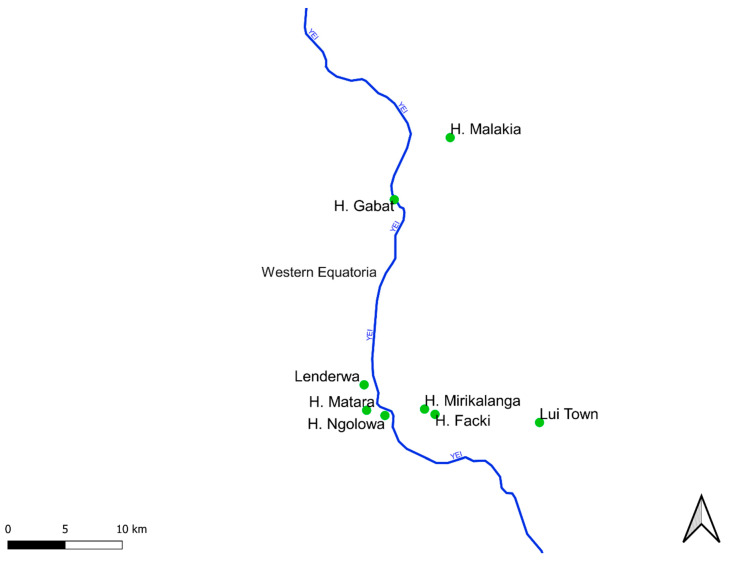
Map showing locations of villages included in the study and their positions relative to the Yei River. These villages were expected to have a high OAE incidence and were, therefore, priority villages for inclusion in a prospective study to demonstrate the impact of an intervention to decrease OAE incidence.

**Figure 3 pathogens-11-00396-f003:**
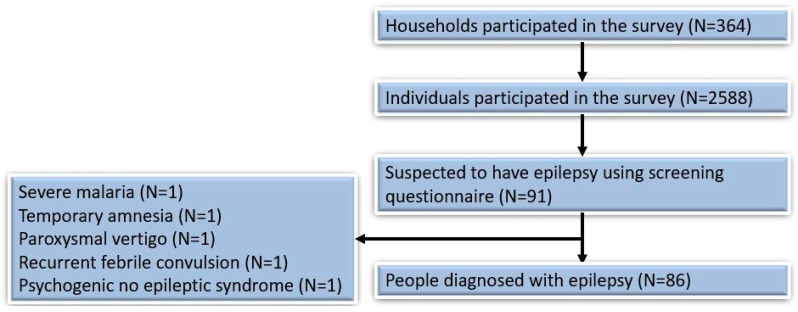
Overview of the survey participants.

**Figure 4 pathogens-11-00396-f004:**
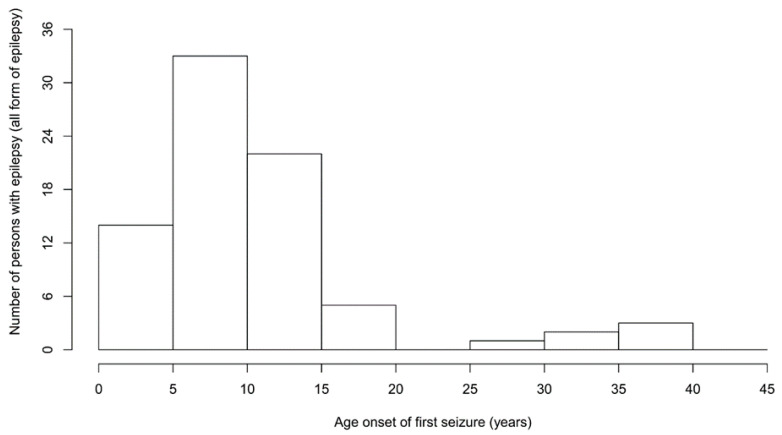
Age of onset of seizures.

**Figure 5 pathogens-11-00396-f005:**
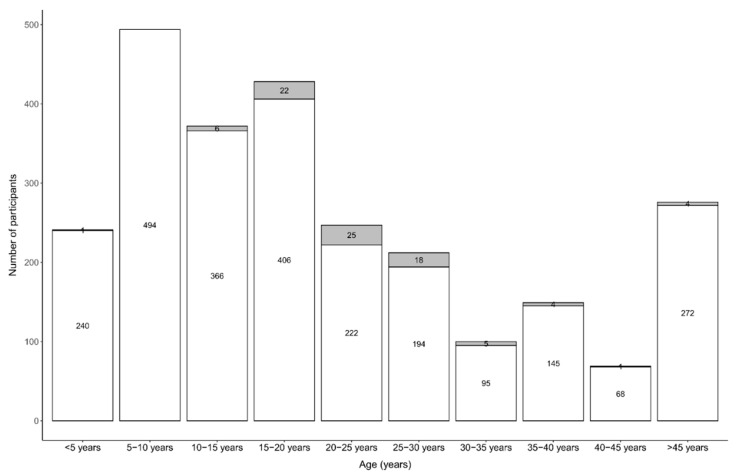
House-to-house survey in Mundri West and East with number of confirmed epilepsy cases per age group, in grey the number of persons with epilepsy.

**Figure 6 pathogens-11-00396-f006:**
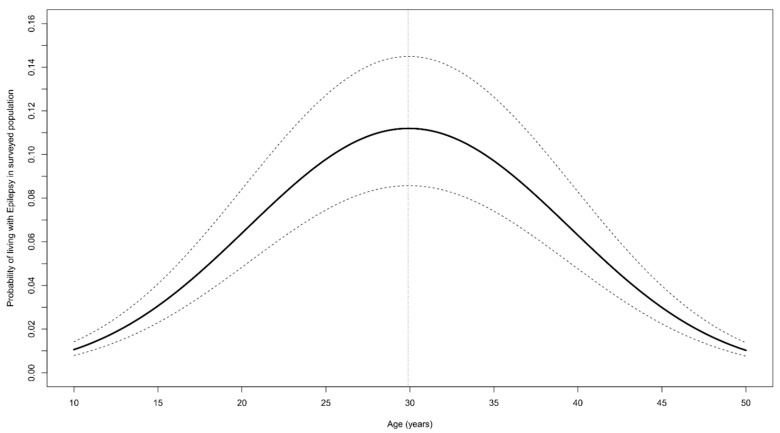
Probability of living with epilepsy according to age (years), estimated from GLMM by keeping other variables as fixed (male person living in a family located close to the Yei River, with income from farming, and with itching and blindness present in the family). The solid black line represent the point estimates and the dashed lines the 95% confidence bands.

**Table 1 pathogens-11-00396-t001:** Number of persons with potential onchocerciasis-related conditions and ivermectin use by study village.

Village	Participants in Survey, n	Epilepsy Confirmed Cases, n (%)	Probable Nodding Syndrome, n (%)	Itching, n (%)	Blindness, n (%)	Ivermectin Use, n (%)	Approximate Distance to Yei River in km
**Mundri West**							
**Amadi Payam**							
Hai Gabat	328	14 (4.3)	4 (1.2)	231 (70.4)	18 (5.5)	155 (47.3)	<2 km
Hai Malakia	256	3 (1.2)	0 (0.0)	174 (67.9)	10 (3.9)	140 (54.7)	>2 km
**Mundri Centre Payam**							
Hai Facki	332	12 (3.6)	3 (0.9)	155 (46.7)	14 (4.2)	213 (64.2)	>2 km
Hai Ngulawa	286	10 (3.5)	4 (1.4)	88 (30.8)	7 (2.4)	136 (47.5)	<2 km
Hai Lenderwa	151	10 (6.6)	4 (2.6)	46 (30.5)	4 (2.6)	85 (56.3)	<2 km
Hai Mirikalanga	372	8 (2.2)	2 (0.5)	133 (35.7)	10 (2.7)	202 (54.3)	>2 km
**Mundri East**							
Lui town	862	28 (3.2)	6 (0.7)	214 (2.5)	7 (0.8)	295 (34.2)	>2 km
**Overall East and West Mundri**	2588	85 (3.3)	23 (0.9)	1041 (40.2)	70 (2.7)	1226 (47.4)	

**Table 2 pathogens-11-00396-t002:** Characteristics of the 86 persons with epilepsy (PWE) examined by clinicians.

Participants’ Characteristics	
Female sex, n (%)	43 (50.0)
Age (years), median (IQR)	25.0 (20.0–29.0)
Born in the study village, n (%)	48 (57.1)
Period (years) of residing in the survey area, median (IQR)	20.0 (10.0–25.0)
Epilepsy features	
Age of onset of the first seizure in all PWE, median (IQR) *	10.0 (8.0–15.0)
Age of onset of the first nodding seizure, median (IQR)	8.0 (5.0–12.0)
Onset of the first seizure last year, n (%)	2 (2.4)
Onset of the first seizure in last 5 years, n (%)	4 (5.0)
Experienced absence(s) or sudden loss of contact with surroundings, for a short duration of time, n (%)	1 (1.2)
Experienced sudden, uncontrollable twitching or shaking of arms, legs, or head, for a period of a few minutes, with amnesia ^ n (%)	76 (90.5)
History of head nodding, n (%)	23 (27.4)
Loss of bladder control, n (%)	58 (69.1)
Foaming at the mouth, n (%)	77 (91.7)
Biting of the tongue, n (%)	68 (80.9)
Most frequent seizure types	
Generalised convulsive seizures only, n (%)	61 (72.6)
Only nodding seizures, n (%)	7 (8.3)
Nodding and convulsive seizures, n (%)	16 (19.1)
Frequency of seizures	
Daily seizure, n (%)	1 (1.2)
Weekly seizure, n (%)	13 (15.5)
Monthly seizure, n (%)	55 (63.1)
Yearly seizure, n (%)	86 (100)
Experienced seizure in the last 12 months, n (%)	72 (83.7)
Seizures/head nodding triggers	
Spontaneous (no obvious trigger), n (%)	67 (79.8)
Sight of food, n (%)	8 (9.5)
Cold weather, n (%)	9 (10.7)
Severe diseases preceding the onset of seizures	
Measles, n (%)	1 (1.2)
Malaria, n (%)	1 (1.2)
Physical examination/symptoms ^&^	
Reduced vision or blind in at least one eye, n (%)	3 (3.6)
Thoracic/spinal abnormalities, n (%)	2 (2.4)
Cervical lymph nodes, n (%)	10 (11.9)
Nakalanga manifestations, n (%)	2 (2.5%)
Itching, n (%)	20 (23.8)
Burn lesions, n (%)	14 (16.7)
Papular/nodular pruritic skin, n (%)	12 (14.3)
Neurological examination ^&^	
Severe cognitive impairment, n (%)	6 (7.1)
Paresis, n (%)	3 (3.6)
Behavioural problem, n (%)	1 (1.2)
Level of autonomy assessed with a modified Rankin scale	
No significant disability despite symptoms (able to carry out all usual duties and activities)	67 (79.8)
Slight disability (unable to carry out all previous activities, but able to look after own affairs without assistance)	8 (9.5)
Moderate disability (requiring some help, but able to walk without assistance)	5 (5.9)
Moderately severe disability (unable to walk without assistance and unable to attend to own bodily needs without assistance)	4 (4.8)
Epilepsy classification	
Epilepsy without head nodding, n (%)	63 (73.2)
Head nodding only, n (%)	7 (8.1)
Head nodding with other seizure types, n (%)	16 (18.6)
Meeting OAE criteria ^+^, n (%)	65 (81.3)
Family members with epilepsy	
Family history of seizures, n (%)	45 (53.6)
Siblings (brother/sister) ^P^, n (%)	37 (82.2)
Father/Mother ^P^, n (%)	7 (15.7)
Grandparent ^P^, n (%)	1 (2.2)
History of anti-seizure medication	
Never used an anti-seizure medication, n (%)	4 (4.6)
Currently taking an anti-seizure medication, n (%)	82 (95.4)
Type of anti-seizure medication ^T^	
Phenobarbital, n (%)	24 (29.4)
Phenytoin, n (%)	1 (1.2)
Carbamazepine, n (%)	55 (67.1)
Sodium valproate, n (%)	1 (1.2)
Ivermectin intake	
Ever received ivermectin, n (%) ^#^	58/74 (78.4)
Ivermectin intake in 2021, n (%) ^$^	33/79 (41.7)
Ivermectin intake in 2021 not known, n (%)	7 (8.9)

n: count; IQR: interquartile range; ^: 2 missing; *: the age onset of the first seizure of 6 PWE was not known; &: 34 missing; T: 4 missing; #: 12 did not remember whether they had ever taken ivermectin; $: 7 PWE did not know their ivermectin intake status. P: the denominator = family with seizure history (n = 45); +: missing information on: age of onset of the first seizure, severe diseases preceding the onset of seizure, or psychomotor problem (n = 7).

**Table 3 pathogens-11-00396-t003:** Characteristics of persons with epilepsy with and without nodding syndrome.

	Epilepsy without Nodding Syndrome (n = 63)	Nodding Syndrome (n = 23)	*p*-Value
Female, n (%)	31 (50.8)	13 (56.5)	0.825
Age (years), median (IQR)	25.0 (20.0–32.0)	24.0 (20.0–28.0)	0.012
Age of onset of first seizures (years)	10.0 (8.0–15.0)	8.0 (5.0–12.0)	0.007
Triggers of seizures			
Sight of food, n (%)	0 (0.0)	8 (34.8)	NA
Cold weather, n (%)	1 (1.6)	7 (30.4)	<0.001
Spontaneous (no obvious trigger), n (%)	60 (95.2)	8 (34.8)	<0.001
Itching, n (%)	13 (20.6)	7 (30.4)	0.556
Burn lesion (s), n (%)	5 (7.9)	5 (21.7)	0.183
Papular/nodular pruritic skin, n (%)	8 (12.7)	4 (17.3)	0.881
Moderate and severe disabilities, n (%)	5 (7.9)	4 (17.4)	0.384

n = number; IQR = Interquartile range; NA = not applicable.

**Table 4 pathogens-11-00396-t004:** Generalised linear mixed model to assess characteristics associated with epilepsy in Mundri West and East.

Characteristics	aOR	95% CI		*p*-Value
Age (years)	1.450	1.286	1.636	<0.0001
Age*Age	0.994	0.992	0.996	<0.0001
Male vs. female sex	1.143	0.712	1.836	0.580
Family income from activities related to the river vs. from other activities ^&^	1.178	0.520	2.670	0.694
Village < 2 km from the Yei River vs. >2 km from the Yei River	1.692	0.918	3.117	0.092
Ivermectin taken during last round vs. no ivermectin taken	0.638	0.376	1.083	0.096
Skin itching vs. no itching in the family	1.014	0.543	1.893	0.965
Blindness/blurred vision vs. no blindness in the family	1.940	0.956	3.940	0.067
Var[b0](se)	1.685 (0.767)			

aOR: adjusted odds ratio; CI: confidence limits; age*age: quadratic effect of age; &: activities related to the river include fishing and agriculture closer to the river, and other activities include employment, trading and military; Var[b0]: variance of random intercept; se: standard error; km = kilometers.

**Table 5 pathogens-11-00396-t005:** Generalised linear mixed model to assess characteristics associated with epilepsy in Mundri West and East (excluding Lui town).

Variables	aOR	95% CI		*p*-Value
Age (years)	1.635	1.353	1.975	<0.0001
Age*Age	0.991	0.988	0.995	<0.0001
Male vs. female sex	0.990	0.546	1.797	0.975
Family income from activities related to the river vs. from other activities ^&^	1.184	0.459	3.054	0.727
Village < 2 km from the Yei River vs. >2 km from the Yei River	2.241	1.034	4.861	0.041
Var[b0](se)	2.419 (1.338)			

aOR: adjusted odds ratio; CI: confidence limits; age*age: quadratic effect of age; &: activities related to the river include fishing and agriculture closer to the river, and other activities include employment, trading and military; Var[b0]: variance of random intercept; se: standard error; km = kilometers.

**Table 6 pathogens-11-00396-t006:** Ivermectin use in the total study population and among persons with epilepsy (PWE) per age group.

Age Group (years)	Ivermectin Use in the Total Study Population, n (%) ^#^	Ivermectin Use Among PWE, n (%) *	*p*-Value ^&^
<5 years	0/240 (0.0)	0 (0.0)	NA
5–10 years	235/492 (48.0)	0 (0.0)	NA
10–15 years	183/369 (49.6)	1/4 (25.0)	0.691
15–20 years	235/426 (55.2)	10/23 (43.5)	0.998
20–25 years	138/245 (56.3)	9/24 (37.5)	0.534
25–30 years	115/211 (54.5)	8/20 (40.0)	0.295
30–35 years	56/99 (56.6)	3/6 (50.0)	0.775
35–40 years	72/149 (48.3)	1/2 (50.0)	0.999
40–45 years	29/69 (42.0)	0/2 (0.0)	NA
>45 years	142/276 (51.4)	1/4 (25.0)	0.584
Overall	1212/2583 (46.9)	33/79 (41.8)	0.706

* denominator = number of PWE in each age group who had taken ivermectin (missing = 7); # denominator = number of survey participants in each age group (missing = 10); &: exact-chi-square test for the differences in ivermectin use among PWE and the total population; NA = not aplicable.

**Table 7 pathogens-11-00396-t007:** Ov16 seroprevalence among children in Amadi and Mundri Centre.

Study Site	Ov16 IgG4 Seroprevalence							Ivermectin Coverage *
	3 years	4 years	5 years	6 years	7 years	8 years	9 years	
Amadi	3/8 (37.5%)	0/4 (0.0%)	3/18 (16.7%)	2/7 (28.5%)	1/11 (9.1%)	4/14 (28.6%)	6/12 (50.0%)	22/62 (35.5%)
Mundri Centre	1/24 (4.2%)	0/17 (0.0%)	1/26 (3.8%)	1/18 (5.5%)	0/26 (0.0%)	1/23 (4.3%)	0/16 (0.0%)	21/109 (19.3%)

* denominator = children eligible for ivermectin (5 years and above); Ov16 IgG4 = *Onchocerca volulus* Immuno globuline G4 antibodies.

## Data Availability

The datasets generated and/or analysed during the current study are publicly accessible via the Zenodo repository (https://doi.org/10.5281/zenodo.6361959, accessed on 20 March 2021).

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
