# Peer review of "The Prevalence of Onchocerciasis-Associated Epilepsy in Mundri West and East Counties, South Sudan: A Door-to-Door Survey"

_pathogens, 2022, doi:10.3390/pathogens11040396_

Round 1

Reviewer 1 Report

In the present manuscript Jada et al describes a cross-sectional study performed in eight study sites in the Greater Mundri area (South Sudan) with the aim to assess the prevalence and predictors of epilepsy in the study region by using an epilepsy survey. They show a high prevalence of epilepsy in the region, but lower than reported in previous studies. The manuscript is clear and in line with similar studies performed in neighbouring regions that are also part of their final prospective study. I have some minor comments and suggestions that can be included/ speculated on in the Discussion, as outlined below. There are also several spelling errors, I listed only some of them below. 

  1. You mention closeness to the fast-flowing river, but more important is the closeness of the study sites to the blackfly breeding sites. Have there been any studies performed to determine the location of breeding sites in the Yei river (or did you investigate this)? Or are you aware of rapids that are more closely located to certain study sites? 

    With that in mind, it might be interesting to consider the geographical location of the households instead of the mere average distance of the village to the river. Have you considered this? 

    That brings me to another point: in Table 1 you define the distance of the study sites to the river as > or < 2km. How did you define this? Blackflies can fly very long distances (10-15km); I feel 2km is not far for them. Also, please define the study sites a bit more in your study design (population size of the village, the exact distance to the river (what is exactly meant with > or <2km?,...). 

    In line with all this, have annual blackfly biting rates been determined in these villages? If not, it might be useful to have an idea of the biting pressure in the distinct study sites (if not now, then for future reference).
  2. Line 147: please elaborate in the text on why 20 years is the cutoff. 

  3. Line 167: Are there no records on ivermectin use in these villages? To get a more accurate depiction?

  4. Figure 5: you mention that the median age for the onset of the first seizure is 10 years, yet in this cross-sectional study you show that epilepsy is most frequent in the age group 20-25 years. Please elaborate on this in the Discussion; is it a result of a high incidence in the past which decreased due to recent reintroduction of CDTI?

    Also, the findings here are different from the other studies performed in Maridi and Mvolo, where e.g. the highest prevalence was observed in the 11-20 age group in Maridi while ivermectin coverage was similar. Can you elaborate/ speculate on this in the Discussion? A higher transmission intensity in Maridi? 

    In line 308-309 you mention that the probability of epilepsy increased with increasing age up to 29.9 years and decreased thereafter, could you explain/ speculate on this in the Discussion? Is it due to high epilepsy-related mortality in this age group, or higher biting pressure in children/ young adults, or are older people less vulnerable,…?

  5. Line 370: What was the prevalence of probable nodding syndrome in Lui town in the present study (as you are now comparing the overall prevalence of this study with the prevalence in Lui town, where the only epilepsy treatment center in the region is located)?

  6. Line 223: did you consider including the different ethnic groups in your GLMM analyses?

  7. Table 2: there is no number given for "Nakalanga manifestations" (2 (2.5%) - according to the text)

  8. Line 297: there is something wrong with this sentence? Please correct.

Some typing errors: 

  • line 93: "Yei" should be "to the Yei river"?
  • line 95: there are two dots at the end of the sentence (the same in Table 4)
  • line 244: in the title of the Table it says "onchocercisis" 
  • line 250: "Table3" change to "Table 3"
  • line 283: "sizures"
  • line 284: "ofthen" 
  • line 324: "excludeding"
  • line 331: "quandratic"
  • line 346: "taken" change to "had taken"
  • line 363: "raison" 
  • line 366: "th" 
  • line 373: "sinds"
  • line 378: "proportions" change to "proportion"
  • line 388: "peronnal" 
  • line 410: "onchocrciasis" 
  • In several places 100,000 is written as 100.000 - please correct this throughout the manuscript

Author Response

Response to

Reviewer 1

In the present manuscript Jada et al describes a cross-sectional study performed in eight study sites in the Greater Mundri area (South Sudan) with the aim to assess the prevalence and predictors of epilepsy in the study region by using an epilepsy survey. They show a high prevalence of epilepsy in the region, but lower than reported in previous studies. The manuscript is clear and in line with similar studies performed in neighbouring regions that are also part of their final prospective study. I have some minor comments and suggestions that can be included/ speculated on in the Discussion, as outlined below. There are also several spelling errors, I listed only some of them below. 

You mention closeness to the fast-flowing river, but more important is the closeness of the study sites to the blackfly breeding sites. Have there been any studies performed to determine the location of breeding sites in the Yei River (or did you investigate this)? Or are you aware of rapids that are more closely located to certain study sites? 

Response

We thank you very much for the very relevant and constructive comments.

We agree it would be important to identify the blackfly breeding sites in the area.

Already twice we submitted a research proposal to carry out an entomological study in the Mundri area, similar to the study we have done in Maridi, but no funding was obtained. However, recently we obtained new funding for research on onchocerciasis-associated epilepsy and we now plan to ask T Lakwo to lead such an entomological investigation in Mundri.

We now include in the conclusion “We also propose to carry out an entomological study in the Greater Mundri area to investigate whether, as in Maridi County, a community based “slash and clear “ vector control method could be implemented.”

With that in mind, it might be interesting to consider the geographical location of the households instead of the mere average distance of the village to the river. Have you considered this? 

Response

We agree it would have been better to have the GPS coordinates of all the households but because of limitation of funding, tablets to collect GPS data were only given to the clinicians who visited households with suspected cases of epilepsy.  

We now state in the discussion “Similar to several other studies in onchocerciasis endemic areas [1, 7, 8], distance of a village to the river was associated with a higher prevalence of epilepsy. However, because of a lack of tablets, we were unable to collect GPS information on all the households included in the study. We also lack information about the localization of the blackfly breading sites and the blackfly biting rates in the study villages.”

That brings me to another point: in Table 1 you define the distance of the study sites to the river as > or < 2km. How did you define this? Blackflies can fly very long distances (10-15km); I feel 2km is not far for them. Also, please define the study sites a bit more in your study design (population size of the village, the exact distance to the river (what is exactly meant with > or <2km?,...). 

Response

We agree that the categorisation of villages > or <2km is not very precise. Given that households in a village are often scattered over a large area, it is difficult to categorise villages by only one GPS.  Therefore, in the absence of GPS data of all households we used the > or <2km classification because in the survey in Maridi, distance to the river <2km had been shown to be a risk factor for epilepsy.

Indeed, blackflies can fly more than 2 km. However, if there are many people living or working on the border of the river, the blackflies will be more likely to bite those people instead of biting people living further away from the river.

In line with all this, have annual blackfly biting rates been determined in these villages? If not, it might be useful to have an idea of the biting pressure in the distinct study sites (if not now, then for future reference).

Response

We agree that it would have been interesting to have blackfly biting rates at the different study sites. This information will also be collected during the planned entomological study.  

  1. Line 147: please elaborate in the text on why 20 years is the cutoff. 

          Response

The 20-year cut-off was chosen because it was considered that in households living since 20 years in the village, most of the children at risk for OAE, would have been born in the village.

  1. Line 167: Are there no records on ivermectin use in these villages? To get a more accurate depiction?

Response

Sorry, we were unable to obtain the ivermectin coverage data per village as reported by the community directed distributors of ivermectin.  

We now state in the discussion “This percentage is much lower than the coverage percentages reported by the community directed distributors (84.7% in Mundri West and 80.2% Mundri East), and the coverage percentages calculated using the National Bureau of Statistics projected population as denominators (83.4% in Mundri West and 70.2% in Mundri East) (MY Logora, personal communication). However, we were unable to compare our ivermectin coverage data with the coverage of data reported by community directed distributors per study village.”

  1. Figure 5: you mention that the median age for the onset of the first seizure is 10 years, yet in this cross-sectional study you show that epilepsy is most frequent in the age group 20-25 years. Please elaborate on this in the Discussion; is it a result of a high incidence in the past which decreased due to recent reintroduction of CDTI?

Also, the findings here are different from the other studies performed in Maridi and Mvolo, where e.g. the highest prevalence was observed in the 11-20 age group in Maridi while ivermectin coverage was similar. Can you elaborate/ speculate on this in the Discussion? A higher transmission intensity in Maridi? 

Response

Thanks for this comment. Indeed, the fact that the epilepsy was most frequent in the 20-25 age group could indeed be a result of a recent decrease in new onset epilepsy in the 10-20 year age group because of the re-introduction of CDTI since 2017.

Indeed, our data suggest the O. volvulus transmission was lower in the Mundri area than in the villages included in the Maridi and Mvolo area. This is also illustrated by the lower OV16 seroprevalence among children 6-9 years old in the Mundri area.

We now state in the discussion: “The highest epilepsy prevalence was observed in the 20−25 years age group. This is in contrast with the findings from similar surveys in Maridi and Mvolo where the highest epilepsy prevalence was observed in the 10−20 years age group. In all these three sites, the ivermectin coverage was low, but the survey in Mundri was done in 2021 after four years of annual CDTI, while in Maridi the survey was done in 2018 after only one CDTI in 2017 [6]. Moreover, in Mundri, the lower OV16 seroprevalence among children below the age of 10 years suggests a lower level of onchocerciasis transmission.”

Response

In line 308-309 you mention that the probability of epilepsy increased with increasing age up to 29.9 years and decreased thereafter, could you explain/ speculate on this in the Discussion? Is it due to high epilepsy-related mortality in this age group, or higher biting pressure in children/ young adults, or are older people less vulnerable,…?

Response

The reason for the decrease of epilepsy after the age of 29.9 is explained by the high mortality of persons with OAE. We now mention in the discussion “The probability of epilepsy increased with increasing age up to 29.9 years and de-creased thereafter. This is explained by a high mortality of persons with OAE below the age of 30 years. Indeed, the median age of death of people with suspected epilepsy was 22 years.”

  1. Line 370: What was the prevalence of probable nodding syndrome in Lui town in the present study (as you are now comparing the overall prevalence of this study with the prevalence in Lui town, where the only epilepsy treatment center in the region is located)?

Response

The prevalence of epilepsy in Lui town was 3.2%; the prevalence of probable nodding syndrome was 0.7%. We have now included the prevalence of nodding syndrome for all the villages.

  1. Line 223: did you consider including the different ethnic groups in your GLMM analyses?

Response

95.1% of the participants belonged to the Moro ethnic group. Therefore, it was not possible to investigate the association between epilepsy and ethnic group

  1. Table 2: there is no number given for "Nakalanga manifestations" (2 (2.5%) - according to the text)

Response

Sorry, we now added this information in the Table

  1. Line 297: there is something wrong with this sentence? Please correct.

            Response

Some typing errors: 

  • line 93: "Yei" should be "to the Yei river"?
  • line 95: there are two dots at the end of the sentence (the same in Table 4)
  • line 244: in the title of the Table it says "onchocercisis" 
  • line 250: "Table3" change to "Table 3"
  • line 283: "sizures"
  • line 284: "ofthen" 
  • line 324: "excludeding"
  • line 331: "quandratic"
  • line 346: "taken" change to "had taken"
  • line 363: "raison" 
  • line 366: "th" 
  • line 373: "sinds"
  • line 378: "proportions" change to "proportion"
  • line 388: "peronnal" 
  • line 410: "onchocrciasis" 
  • In several places 100,000 is written as 100.000 - please correct this throughout the manuscript

           Response

Sorry for these typing errors. They now have been corrected. We also corrected some references.

Reviewer 2 Report

General comment

In this article, the authors conducted a baseline survey in onchocerciasis-endemic villages in South-Sudan to assess the prevalence of epilepsy in the communities.   They confirmed the close relationship with both clinical and infection entities as well as the low coverage of preventive therapy for onchocerciasis.

The relevance of this MS is to bring robust data allowing to design a more appropriate control intervention against onchocerciasis, which may include more than one annual ivermectin distribution. Overall, the defined methodology and statistics seems quite adequate and clearly described; the manuscript is well written and clear to the readers

Author Response

Reviewer 2

In this article, the authors conducted a baseline survey in onchocerciasis-endemic villages in South-Sudan to assess the prevalence of epilepsy in the communities.   They confirmed the close relationship with both clinical and infection entities as well as the low coverage of preventive therapy for onchocerciasis.

The relevance of this MS is to bring robust data allowing to design a more appropriate control intervention against onchocerciasis, which may include more than one annual ivermectin distribution. Overall, the defined methodology and statistics seems quite adequate and clearly described; the manuscript is well written and clear to the readers

Response

Thanks